# Current Status and Future Perspective on the Management of Lymph Node-Positive Prostate Cancer after Radical Prostatectomy

**DOI:** 10.3390/cancers14112696

**Published:** 2022-05-30

**Authors:** Masaki Shiota, Leandro Blas, Masatoshi Eto

**Affiliations:** Department of Urology, Graduate School of Medical Sciences, Kyushu University, Fukuoka 812-8582, Japan; leandro.blas85@hotmail.com.ar (L.B.); eto.masatoshi.717@m.kyushu-u.ac.jp (M.E.)

**Keywords:** androgen deprivation therapy, lymph node involvement, pelvic lymph node dissection, radiotherapy, radical prostatectomy

## Abstract

**Simple Summary:**

pN1 after RP with PLND represents one of the most unfavorable prognostic factors for disease recurrence and cancer-specific mortality in prostate cancer. Treatment intensification may reduce risks of recurrence and cancer-specific mortality, but it may increase adverse events and impair quality of life. However, optimal management for pN1 patients remains unclear. Nevertheless, few randomized control trials for pN1 are under investigation, and then more research is needed to establish an optimal therapeutic strategy for patients with pN1. This review summarizes current evidence on the treatments available for men with pN1, summarizes RCTs that included pN1 prostate cancer, and also discusses future perspectives.

**Abstract:**

Pathological lymph node involvement (pN1) after a pelvic lymph node dissection represents one of the most unfavorable prognostic factors for disease recurrence and cancer-specific mortality in prostate cancer. However, optimal management for pN1 patients remains unclear. Thus, the guideline from the European Association of Urology recommends discussing three following management options with pN1 patients after an extended pelvic lymph node dissection, based on nodal involvement characteristics: (i) offer adjuvant androgen-deprivation therapy, (ii) offer adjuvant androgen-deprivation therapy with additional radiotherapy and (iii) offer observation (expectant management) to a patient with ≤2 nodes and a prostate-specific antigen <0.1 ng/mL. Treatment intensification may reduce risks of recurrence and cancer-specific mortality, but it may increase adverse events and impair quality of life. Few randomized control trials for pN1 are under investigation. In addition, there are limited reports on the quality of life and patient-reported outcomes in patients with pN1. Therefore, more research is needed to establish an optimal therapeutic strategy for patients with pN1. This review summarizes current evidence on the treatments available for men with pN1, summarizes randomized control trials that included pN1 prostate cancer, and discusses future perspectives.

## 1. Introduction

Pelvic lymph node dissection (PLND) is recommended during radical prostatectomy (RP) for prostate cancer in clinical practice, depending on risk classification [1]. PLND is the gold standard procedure for the diagnosis of lymph node involvement (LNI), although so far, its therapeutic value has not been proven [2,3]. Pathological LNI (pN1) rates after RP with PLND vary between 0% and 37% depending on risk classification and the areas removed in PLND [4]. LNI represents one of the most unfavorable prognostic factors for recurrence and cancer-specific mortality [5].

So far, the only randomized clinical trial (RCT) performed for patients solely with pN1 prostate cancer showed that immediate androgen deprivation therapy (ADT) was associated with better overall survival (OS) than deferred ADT in patients with distant metastases or symptomatic recurrences [6]. However, this finding cannot be generalized to all patients with pN1. First, this study started in the pre-prostate specific antigen (PSA) era, and limited PLND was performed, which is no longer a standard practice [1]. Nevertheless, the median number of positive lymph nodes removed was higher than in recent studies [6]. Second, the initiation of deferred ADT may be delayed too long, as early ADT should be reserved for those men at the highest risk of disease progression and a long-life expectancy [1]. Therefore, it remains an open question whether the prognosis of early salvage ADT can be equivalent to immediate ADT. It has been shown that the survival between observation and adjuvant ADT was comparable using the Surveillance, Epidemiology, and End Results database [7]. In addition, several retrospective studies have suggested that the long-term prognosis in pN1 patients is heterogeneous and varies according to disease characteristics, such as the number of positive nodes, disease extension, margin status in RP, and PSA kinetics [8,9]. Meanwhile, favorable disease control and better survival by the addition of radiation therapy (RT) to immediate ADT have been reported by retrospective studies. Thus, RT plus ADT appeared to be a promising approach to improve the prognosis among men with pN1 prostate cancer. However, given the lack of level-1 evidence applicable to contemporary patients, the European Association of Urology (EAU) has recommended practitioners to discuss with pN1 patients three management options after an extended PLND, based on nodal involvement characteristics: (i) offer adjuvant ADT, (ii) offer adjuvant ADT with additional RT and (iii) offer observation (expectant management) to a patient with ≤2 nodes and a PSA < 0.1 ng/mL after extended PLND [1].

Clinical evidence on optimal treatment for pN1 prostate cancer is limited, requiring the establishment of a treatment strategy since treatment outcomes for pN1 vary from satisfactory, with observation in patients with favorable characteristics, to unsatisfactory, even with intensive treatment in patients with unfavorable characteristics. This review summaries current evidence on the treatments available for men with pN1, RCTs that included pN1 prostate cancer, and discussed future perspectives.

## 2. The Prognosis in pN1 Prostate Cancer by Treatments

Several retrospective studies reported the prognosis in pN1 prostate cancer (Table 1). Since the standard treatment for pN1 has not been established, management strategies differed among studies. Biochemical recurrence (BCR)-free survival rate is affected by adjuvant therapy and varies from 28% to 61% at five years. Recurrence-free survival (RFS), determined basically by radiological recurrence, and metastasis-free survival (MFS) was 55–84% and 65–80% at 10 years, respectively. Cancer-specific survival (CSS) and OS were ~80% and ~70% at 10 years, respectively. Although survival in patients with pathological negative LNI (pN0) or unknown LNI (pNx) after RP is generally excellent, the prognosis in pN1 prostate cancer is inferior, making the improvement of treatment outcomes an unmet need, where treatment intensification is an attractive approach.

Interestingly, Moschini et al. have reported the natural history of clinical recurrence patterns of pN1 prostate cancer using a database from the Mayo Clinic, where almost all patients received adjuvant or salvage ADT, and 22% received adjuvant or salvage RT [9]. In that study, 17% and 28% of patients presented clinical recurrence in soft local tissues and pelvic lymph nodes, respectively [9]. Similarly, prostate-specific membrane antigen (PSMA) positron emission tomography (PET)/computed tomography (CT) detected positive lesions only in the pelvic area in 65%, only outside the pelvic area in 10%, and both in 25% of patients [22]. Those findings on the locations of recurrent disease in pN1 prostate cancer suggest that loco-regional disease control after RP is feasible in a subset of patients.

Table 2 shows retrospective studies and database studies on the prognostic differences with the addition of RT. Tilki et al. and Da Pozzo et al. have reported favorable BCR-free survival and MFS with adjuvant RT than those men without adjuvant RT, using large case series from a European institution [10,23]. Similarly, it has recently been reported that RT for persistent PSA after RP was associated with improved RFS and MFS in large case series from Japanese multiple institutions [19]. Consistently, several retrospective studies from high-volume centers in the United States (US) and Europe have reported that RT plus adjuvant ADT was associated with better CSS and OS [15,16,23,24]. Similarly, other studies using a national database in the US have demonstrated that patients treated with adjuvant RT were associated with better CSS and OS than those without adjuvant RT [25,26,27,28]. Thus, the addition of RT to immediate ADT may result in favorable disease control and better survival. However, careful interpretation is necessary because retrospective studies suffer from potential biases, such as treatment selection. Thus, it is necessary to examine the value of RT for pN1 prostate cancer in RCTs.

Treatment de-intensification is another approach for pN1 patients with a favorable prognosis. Tilki et al. had demonstrated that BCR-free survival and MFS were better in men who received adjuvant RT compared with no treatment or salvage RT using large case series data from Martini-Klinik [10]. Moreover, they have recently reported a higher CSS with adjuvant RT than with salvage RT [30]. Consistently, expectant management for pN1 was associated with inferior survival than adjuvant treatment including RT in retrospective case series and database studies [24,27,28,31]. In contrast, the RADICALS-RT trial that enrolled mainly pN0 or pNx in addition to pN1 patients has recently reported a comparable BCR rate between adjuvant RT and salvage RT [32]. However, this RCT included only 5% of patients with pN1, limiting the applicability of the findings [32]. Based on these studies, the EAU guideline recommends observation in limited patients with favorable prognosis features [1]. Thus, stratification by patient’s and tumor’s characteristics may be a promising approach.

## 3. Treatment Burden in pN1 Patients

RP with PLND could cause various postoperative complications such as urinary incontinence, erectile dysfunction, bladder neck contracture, and inguinal hernia, reducing the quality of life (QoL). The Prostate Testing for Cancer and Treatment (ProtecT) trial, which randomized patients to receive monitoring, RP or RT, found the most significant negative effect on urinary continence and sexual function in those patients undergoing RP [33]. Similarly, a prospective trial showed that patients who underwent RP had worse urinary incontinence and a worse sexual domain score compared with patients with RT or active surveillance [34]. Thus, additional treatment after RP may lead to further deterioration in QoL.

Additionally, ADT can cause several adverse effects (AEs) including sexual dysfunction, hot flushes, bone fractures, metabolic effects, cardiovascular morbidity, fatigue, and neurological disorders [35]. A prospective observational study that included patients with locally advanced prostate cancer or PSA relapse after local therapy found that immediate ADT was associated with a lower overall QoL than in those with deferred treatment [36]. Consistently, in another prospective observational study, patients undergoing ADT, after RP or RT, showed higher levels of depression, worse self-body image perception, worse sleep quality, and worse QoL than controls [37].

Similarly, salvage RT is also associated with toxicity. A prospective study evaluating salvage RT plus ADT after RP showed increased bowel dysfunction and urinary dysfunction by the end of RT. These rates improved after RT completion, but not completely. Meanwhile, erectile function presented no change during RT but showed an abrupt decline after RP [38]. Similarly, in an observational study from the Martini-Klinik Prostate Cancer Center, patients who received RT after RP had a higher incontinence rate and lower potency rate than matched RP-only patients. Both rates increased further with the addition of ADT [39]. Thus, currently available data on toxicity demonstrate an increased incidence of acute and long-term grade 2 AEs and transient decline in QoL outcomes, but no significant increase in long-term grade 3–4 AEs with the use of RT after RP [40].

Based on this evidence, treatment addition after RP may lead to increased toxicity and reduced QoL. Meanwhile, treatment addition may reduce or delay recurrence, which may lead to a recovery of QoL by avoiding continuous ADT. However, there is little data on QoL and outcomes reported by patients with pN1. Thus, a prospective study on treatment strategies would be necessary.

## 4. RCTs That Include pN1 Prostate Cancer

So far, various RCTs have been conducted to develop adjuvant therapy for high risk of recurrence and salvage therapy for BCR after RP. Due to the unfavorable prognosis among men with pN1 prostate cancer, the development of novel treatment to suppress recurrence and improve survival is necessary. However, the frequency of pN1 prostate cancer is low, and so far, only the RCT by Messing et al. enrolled patients with pN1 only [6]. Moreover, a few RCTs included pN1 patients in addition to pN0 or pNx (Table 3). Among them, TAX3501 (NCT00283062) and TAX3503 (NCT00514917) trials examined the efficacy of docetaxel plus ADT after RP regardless of LNI in adjuvant or salvage setting. However, these RCTs showed no significant improvement in biochemical or radiographic progression-free survival [41,42]. Similarly, the SWOG S9921 (NCT00004124) trial demonstrated no benefit in CSS and OS with the addition of mitoxantrone [17]. Meanwhile, the GETUG-AFU 12 trial included patients with high-risk nonmetastatic prostate cancer (stage T3-T4 disease, Gleason score of ≥8, PSA >20 ng/mL, or pN1) who underwent staging PLND without RP showed longer relapse-free survival with ADT plus docetaxel and estramustine than with ADT alone [43]. However, as patients included in this trial never underwent RP, it is not appropriate to apply this finding to patients with pN1 after RP and PLND.

To date (31 March 2022), several RCTs recruiting patients with pN1 are being performed (Table 4). Among them, two RCTs are recruiting patients solely with pN1. The INNOVATE NRG-GU008 (NCT04134260) trial is recruiting 586 patients with pN1 prostate cancer after RP, to evaluate the effect of apalutamide for two years on MFS and ADT plus RT to pelvis and prostate bed in an adjuvant setting. This RCT may expand the indication of a novel androgen receptor pathway inhibitor, apalutamide, to pN1 disease in case of a positive result. The PROPER (NCT02745587) trial is also recruiting solely pN1 patients. This trial compares the presence of clinical relapse (loco-regional recurrence or distant metastases) between RT to the prostate bed alone or in combination with the pelvic lymph node regions, plus two years of ADT in an adjuvant setting. The result of this RCT will be important to determine the appropriate field of RT for pN1 patients.

Moreover, six RCTs recruiting pN1 in addition to pN0 or pNx are under investigation (Table 3). However, these RCTs do not aim to develop the treatment specific for pN1. Actually, the reported standard treatments for pN1 patients are different across trials, making the generalization of results difficult.

In the future, ongoing RCTs may change the current therapeutic landscape in pN1 prostate cancer. As examined in the PATRON (NCT04557501) trial, PSMA PET/CT is a promising modality that has the potential to change the landscape of treatment strategies for prostate cancer. Recently, proPSMA Study Group Collaborators have reported that PSMA PET/CT showed higher accuracy for pelvic nodal metastases and distant metastases than traditional imaging modalities [44]. Despite excellent specificity, PSMA PET/CT presents a lower sensitivity (of ~40%) for detection of positive pelvic lymph nodes than extended PLND [45,46,47]. This suggests that PSMA PET/CT alone cannot yet replace the diagnostic role of extended PLND, although combination with nomograms was suggested to improve predictive ability [48]. Meanwhile, PSMA PET/CT has shown good potential in patients with BCR after RP, where detection rates increase according to PSA level: 33% (95% confidence interval (CI), 16–51%) in PSA < 0.2 ng/mL, 45% (95% CI, 39–52%) in PSA 0.2–0.49 ng/mL, 59% (95% CI, 50–68%) in PSA 0.5–0.99 ng/mL, 75% (CI: 66–84%) in PSA 1.0–1.99 ng/mL, and 95% (95% CI, 92–97%) in PSA ≥2.0 ng/mL [49]. Thus, PSMA PET/CT may be useful for patients with BCR. However, the residual tumor is not detectable in most patients without BCR after RP, and a conventional therapeutic approach is necessary for those patients with no visible lesion. In addition, it is unknown whether a more sensitive diagnosis by PSMA PET/CT leads to improved outcomes, which will be addressed by the PATRON trial.

## 5. Conclusions

In conclusion, pN1 represents an unfavorable characteristic in patients who underwent RP with PLND. Currently, a gold standard strategy for pN1 has not been established, and survival of patients with pN1 is not satisfactory. Thus, the development of novel treatments with more efficacy and less toxicity is an unmet need. Treatment intensification by adding novel antiandrogen as well as treatment guided by next-generation imaging are currently under investigation, and may become a standard of care in the future. However, a few RCTs are underway. In addition, as studies on QoL and patient-reported outcomes in patients with pN1 are scarce, more research on QoL and patient-reported outcomes is needed to develop a better therapeutic strategy for pN1 patients.

## Figures and Tables

**Table 1 cancers-14-02696-t001:** Prognosis among men with pN1.

**Authors**	* **n** *	**Groups**	**Median Follow-Up**	**Time (year)**	**BCR-Free Survival (%)**	**Reference**
Tilki et al.	773	All	33.8 (month)	4	43.3	[10]
		Matched pair cohorts	–	–	–	
	192	Observation		4	43	
	192	aRT		4	57	
Fleischmann et al.	102	Observation	7.7 (year)	5	28	[11,12]
Touijer et al.	369	Observation	4 (year)	10	28	[8]
Dorin et al.	150	All	10.4 (year)	10	57	[13]
	49	Observation	11.4 (year)	10	59	
Hofer et al.	201	aADT	41 (month)	5	61	[14]
Abdollah et al.	1107	aADT/aRT	7.1 (year)	10	56	[15,16]
**Authors**	* **n** *	**Groups**	**Median follow-up**	**Time (year)**	**RFS (%)**	**Reference**
Hussain et al.			11.2 (year)	–	–	[17]
	79	aADT		10	55	
	83	aADT + mitoxantrone and prednisone		10	66	
Bravi et al.			77(month)	10		[18]
	100	aRT	–		92	
	272	aADT + aRT	–		70	
Dorin et al.	150	All	10.4 (year)	10	84	[13]
	49	Observation	11.4 (year)	10	80	
Shiota et al.	561	All	4.8 (year)	510	8775	[19]
**Authors**	* **n** *	**Groups**	**Median follow-up**	**Time (year)**	**MFS (%)**	**Reference**
Tilki et al.	773	All	33.8 (month)	4	86.6	[10]
		Matched pair cohorts	–			
	192	Observation		4	82.5	
	192	aRT			91.8	
Touijer et al.	369	Observation	4 (year)	10	65	[8]
Shiota et al.	561	All	4.8 (year)	510	9080	[19]
**Authors**	* **n** *	**Groups**	**Median follow-up**	**Time (year)**	**CSS (%)**	**Reference**
Bravi et al.			77 (month)	10	–	[18]
	100	aRT			98	
	272	aADT + aRT			92	
Mandel et al.	209	Observation	60.2 (month)			[20]
Fleischmann et al.	102	Observation	7.7 (year)	5	78	[11,12]
Touijer et al.	369	Observation	4 (year)	10	72	[8]
Abdollah et al.	1107	aADT/aRT	7.1 (year)	10	83.6	[15,16]
Bianchi et al.	518	aADT/aRT	52 (month)	8	71.2	[21]
Shiota et al.	561	All	4.8 (year)	510	9891	[19]
**Authors**	* **n** *	**Groups**	**Median follow-up**	**Time (year)**	**OS (%)**	**Reference**
Hussain et al.			11.2 (year)	–	–	[17]
	79	aADT		10	81	
	83	aADT + mitoxantrone and prednisone		10	81	
Bravi et al.			77 (month)	10		[18]
	100	aRT			81	
	272	aADT + aRT			85	
Fleischmann et al.	102	Observation	7.7 (year)	5	75	[11,12]
Touijer et al.	369	Observation	4 (year)	10	60	[8]
Dorin et al.	150	All	10.4 (year)	10	74	[13]
	49	Observation	11.4 (year)	10	81	
Abdollah et al.	1107	aADT/aRT	7.1 (year)	8	78.1	[15,16]
Shiota et al.	561	All	4.8 (year)	510	9789	[19]

aADT, adjuvant androgen deprivation therapy; aRT, adjuvant radiotherapy; BCR, biochemical recurrence; CSS, cancer-specific survival; OS, overall survival; RFS, recurrence-free survival; MFS, metastasis-free survival.

**Table 2 cancers-14-02696-t002:** Hazard ratio for prognosis by multivariate analysis or propensity score matched analysis in pN1 cases.

**Authors**	**Accrual Years**	**Cohort**	**Treatment**	* **n** *	**BCR-Free Survival**	***p*-Value**	**Reference**
Tilki et al.	2005–2013	Martini-Klinik	aRT	213	Ref		[10]
			aADT	55	2.14 (1.33–3.45)	0.002	
			Observation	505	2.22 (1.61–3.13)	<0.001	
Da Pozzo et al.	1988–2002	Vita-Salute San Raffaele University	aADT	121	Ref		[23]
			aADT + aRT	129	0.49	0.002	
**Authors**	**Accrual years**	**Cohort**	**Treatment**	* **n** *	**RFS**	* **p** * **-value**	
Bravi et al.	1991–2017	Vita-Salute San Raffaele University	aRT	100	Ref		[18]
			aADT + aRT	272	2.41 (1.09–5.31)	0.029	
Shiota et al.	2006–2019	Multicenter in Japan	aADT	188	Ref		[19]
			aRT	24	4.42 (2.42–8.07)	<0.0001	
			aADT + aRT	58	0.42 (0.21–0.82)	0.011	
**Authors**	**Accrual years**	**Cohort**	**Treatment**	* **n** *	**MFS**	* **p** * **-value**	
Tilki et al.	2005–2013	Martini-Klinik	aRT	213	Ref		[10]
			aADT	55	2.81 (1.60–4.92)	0.014	
			Observation	505	2.78 (1.61–5.00)	<0.001	
Shiota et al.	2006–2019	Multicenter in Japan	aADT	188	Ref		[19]
			aRT	24	1.67 (0.67–4.16)	0.27	
			aADT + aRT	58	0.37 (0.15–0.93)	0.034	
**Authors**	**Accrual years**	**Cohort**	**Treatment**	* **n** *	**CSS**	* **p** * **-value**	
Wong YN et al.	1991–1999	SEER	Observation	522	Ref		[7]
			aADT	209	0.97 (0.56–1.68)	NA	
Kim et al.	2004–2014	SEER	No aRT	905	Ref		[25]
			aRT	905	0.63 (0.44–0.88)	NA	
Da Pozzo et al.	1988–2002	Vita-Salute San Raffaele University	aADT	121	Ref		[23]
			aADT + aRT	129	0.38	0.009	
Abdollah et al.	1988–2010	Mayo ClinicVita-Salute San Raffaele University	aADT + aRT	386	Ref		[15]
			aADT	721	2.72 (1.62–4.55)	<0.001	
Briganti et al.	1988–2003	Mayo ClinicVita-Salute San Raffaele University	aADT + aRT	117	Ref		[29]
			aADT	247	2.5	0.004	
Bravi et al.	1991–2017	Vita-Salute San Raffaele University	aRT	100	Ref		[18]
			aADT + aRT	272	5.39 (0.70–41.39)	0.11	
Touijer et al.	1988–2010	Memorial Sloan Kettering Cancer CenterMayo ClinicVita-Salute San Raffaele University	Observation	387	Ref		[24]
			aADT	676	0.64 (0.43–0.95)	0.027	
			aADT + aRT	325	0.26 (0.15–0.44)	<0.0001	
Tilki et al.	1995–2017	Martini-Klinik	sRT	3040	Ref		[30]
			aRT	851	0.92 (0.85–0.99)	0.03	
**Authors**	**Accrual years**	**Cohort**	**Treatment**	* **n** *	**OS**	* **p** * **-value**	
Zareba et al.	2004–2010	NCDB	Observation	4889	Ref		[31]
			aADT	1571	1.06 (0.87–1.29)	0.56	
			aRT	355	0.75 (0.50–1.10)	0.14	
			aADT + aRT	976	0.69 (0.52–0.92)	0.010	
Jegadeesh et al.	2003–2011	NCDB	aADT + aRT	906	Ref		[26]
			aADT	1663	1.50 (1.18–1.90)	<0.001	
Wong AT et al.	2004–2011	NCDB	Observation	3636	Ref		[27]
			aADT	2041	0.99 (0.85–1.15)	0.90	
			aRT	350	1.02 (0.74–1.40)	0.92	
			aADT + aRT	1198	0.67 (0.55–0.83)	<0.001	
Gupta et al.	2004–2013	NCDB	Observation	4489	Ref		[28]
			aADT	2065	1.01 (0.87–1.18)	0.88	
			aADT + aRT	1520	0.77 (0.64–0.94)	0.008	
Wong YN et al.	1991–1999	SEER	Observation	522	Ref		[7]
			aADT	209	0.95 (0.71–1.27)	NA	
Abdollah et al.	1988–2010	Mayo ClinicVita-Salute San Raffaele University	aADT + aRT	386	Ref		15
			aADT	721	2.08 (1.41–3.05)	<0.001	
Briganti et al.	1988–2003	Mayo ClinicVita-Salute San Raffaele University	aADT + aRT	117	Ref		[29]
			aADT	247	2.3	<0.001	
Bravi et al.	1991–2017	Vita-Salute San Raffaele University	aRT	100	Ref		[18]
			aADT + aRT	272	0.91 (0.45–1.84)	0.8	
Touijer et al.	1988–2010	Memorial Sloan Kettering Cancer CenterMayo ClinicVita-Salute San Raffaele University	Observation	387	Ref		[24]
			aADT	676	0.90 (0.65–1.25)	0.5	
			aADT + aRT	325	0.41 (0.27–0.64)	<0.0001	

aADT, adjuvant androgen deprivation therapy; aRT, adjuvant radiotherapy; BCR, biochemical recurrence; CSS, cancer-specific survival; OS, overall survival; MFS, metastasis-free survival; NA, not available; NCDB, National Cancer Database; RFS, recurrence-free survival; SEER, Surveillance, Epidemiology, and End Results; sRT, salvage radiotherapy.

**Table 3 cancers-14-02696-t003:** Phase 3 randomized clinical trials for pN1 prostate cancer after RP.

Clinical Trial ID	Trial Name	Investigator	pN Status	Curative Treatment	Patients Number	PSA Criteria for Inclusion	Intervention Timing	Standard of Care (Trial Treatment)	Trial Treatment 1	Primary Endpoint	Follow-Up Period	Result	Reference
-	EST 3886	Eastern Cooperative Oncology Group study	pN1	RP	98	Not defined	Adjuvant/salvage treatment	Immediate goserelin or castration	Salvage goserelin or castration when clinical recurrence	Clinical recurrence-free survival	Median, 11.9 years	Positive	[6]
NCT00541047	RADICALS-RT	Medical Research Council	pN0/Nx/N1	RP	1396	Undetectable PSA (≤0.2 ng/mL)	Adjuvant/salvage treatment	Immediate RT (prostate bed + pelvic LN) ± hormone therapy	Salvage RT with or without hormone therapy	Metastasis-free survival	Median, 4.9 years	Negative	[32]
NCT00283062	TAX3501	Sanofi	pN0/N1	RP	228	Undetectable PSA (≤0.2 ng/mL)	Adjuvant treatment	Adjuvant or salvage leuprolide (18 months)	SOC plus docetaxel for 6 cycles	Progression-free survival (PSA progression, radiological, or histological progression)	Median, 3.4 years	Negative	[41]
NCT00514917	TAX3503	Sanofi	pN0/Nx/N1	RP	413	Elevated PSA (≥1 ng/mL)	Salvage treatment	Leuprolide (up to 18 months) plus 4-week bicalutamide	SOC plus docetaxel for up to 10 cycles	Progression-free survival (PSA progression, or radiological progression)	Median, 2.8 years	Negative	[42]
NCT00004124	SWOG S9921	Southwest Oncology Group	pN0/Nx/N1	RP	983	Undetectable PSA (≤0.2 ng/mL)	Adjuvant treatment	Goserelin plus bicalutamide	SOC plus mitoxantrone and prednisone	Overall Survival and disease-specific survival	Median, 11.2 years	Negative	[17]
NCT00765479	CDR0000615902	University of Illinois at Chicago	pN0/Nx/N1	RP	284	Undetectable PSA (<0.07 ng/mL)	Adjuvant treatment	Casein placebo beverage	Soy protein isolate beverage	Two-year PSA failure rate and Time to PSA failure	2 years	Negative	[50]

LN, lymph node; PSA, prostate-specific antigen; RP, radical prostatectomy; RT, radiotherapy; SOC, standard of care.

**Table 4 cancers-14-02696-t004:** Phase 3 randomized clinical trials under investigation for node-positive prostate cancer after RP.

Clinical Trial ID	Trial Name	Investigator	pN Status	Curative Treatment	Patients Number	PSA Criteria for Inclusion	Intervention Timing	Standard of Care	Trial Treatment 1	Trial Treatment 2	Primary Endpoint	Follow-Up Period	Study Start Date	Estimated Completion Date
NCT04134260	INNOVATE NRG-GU008	NRG Oncology	pN1	RP	586	PSA (>0 ng/mL)	Adjuvant treatment	Hormone therapy (24 months) plus RT (prostate bed and pelvis)	SOC plus apalutamide (720 days)		Metastasis-free survival	90 months	Mar-20	Nov-26
NCT02745587	PROPER	University Hospital, Ghent	pN1	RP	330	Not defined	Adjuvant treatment	RT (prostate bed and pelvis) plus ADT (2 years)	RT (prostate bed) plus ADT (2 years)		Clinical recurrence presence of loco-regional relapse or distant metastases	96 months	Apr-16	Apr-21
NCT01442246	GETUG-AFU-20	UNICANCER	pN0/Nx/N1	RP	700	PSA (<0.1 ng/mL)	Adjuvant treatment	Observation	Leuprolide (24 months)		Metastasis-free survival	120 months	Jul-11	Sep-27
NCT00541047	RADICALS-HD	Medical Research Council	pN0/Nx/N1	RP	4236	PSA (≤5 ng/mL)	Adjuvant/salvage treatment	RT alone	RT plus hormone therapy (6 months)	RT plus hormone therapy (2 years)	Disease-specific survival (i.e., death due to prostate cancer)	84 months	Nov-07	Sep-21
NCT03119857	SPCG-14	Scandinavian Prostate Cancer Group	pN0/Nx/N1	RP/RT	349	Elevated PSA †	Salvage treatment	Antiandrogen (bicalutamide)	SOC plus docetaxel (up to 8–10 cycles)		Progression free survival (PSA progression or radiographic progression)	60 months	Feb-09	Apr-23
NCT02319837	EMBARK	Pfizer	pN0/Nx/N1	RP/RT	1068	Elevated PSA ‡	Salvage treatment	Placebo plus leuprolide	Enzalutamide monotherapy	Enzalutamide plus leuprolide	Metastasis-free survival	Approximately 90 months	Dec-14	Aug-22
NCT03009981	AFT-19	Alliance Foundation Trials, LLC.	pN0/Nx/N1	RP	504	PSA (>0.5 ng/mL)	Salvage treatment	Degarelix monotherapy or leuprolide plus bicalutamide	SOC plus apalutamide (52 weeks)	SOC plus apalutamide and abiraterone acetate (52 weeks)	PSA progression-free survival	30 months	Mar-17	Jan-23
NCT04557501	PATRON	CHUM	pN0/Nx/N1	RP	776	PSA (>0.1 ng/mL)	Salvage treatment	Treatment without PSMA PET/CT	PSMA PET/CT guided intensification of therapy		Failure-free survival (PSA or radiographic recurrence)	60 months	Jan-21	Oct-28

PSA, prostate-specific antigen; RP, radical prostatectomy; RT, radiotherapy; SOC, standard of care. † >10 ng/mL or PSA-doubling time < 12 months and PSA > 0.5 ng/mL after RP, and PSA > +2.0 ng/mL above nadir and PSA > 10 ng/mL or PSA-doubling time < 12 months and PSA > 0.5 ng/mL after RT. ‡ >1.0 ng/mL after RP and >2.0 ng/mL above nadir after RT.

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
