# Peer review of "Current Status and Future Perspective on the Management of Lymph Node-Positive Prostate Cancer after Radical Prostatectomy"

_cancers, 2022, doi:10.3390/cancers14112696_

Round 1

Reviewer 1 Report

The review article by Shiota et al, focused on Current status and future perspective on the management of 2 lymph node-positive prostate cancer after radical prostatectomy covers the treatment available options for prostate cancer patients with pN1, RCTs that includes pN1 prostate cancer.

It true that multimodality treatment offers promising results in patients with histopathological pN1 prostate cancer because pN1 has poor prognosis

The author did mention the current management however the future management is not well described in the article.

In section “treatment burden in pN1 patients’’ how the cancer stage or grade is critical should also be mentioned which was lacking in the article.

The reference needs to add more updated articles from recent years, of the year 2021 only 2 papers of were added in the article.

Overall, the article is well written and still need to mention some of the important factor involved in characterizing the patients or categorizing the patients. In particular which age group of patients and grade cancer is suitable for the mentioned treatment option for prostate cancer patients with pN1.

Reviewer 2 Report

In this review article, Shiota et al. are summarizing the current knowledge on the management of lymph node-positive prostate cancer after radical prostatectomy.

General comment:

The topic of the article is highly relevant. The article is written in a concise way but it covers all the important issues. It is a valuable source of information and considerations on the topic.

Specific points:

Table 1: Since there is a repetition of information to a certain extent (the same studies searched for BCR-free survival, RFS, MFS, CSS or OS) would it be possible to rearrange the Table so that the information from the same study is not repeated? This is only suggestion, I leave this up to the authors to decide which way of representing the data would be the best.

Table 1: 'Median follow up'; it is not indicated for all values whether it is in months (mo), or years (yr).

Line 44: 'differed' instead of 'deferred'

Lines 79-80: 'pN0' and 'pNx' should be explained

Line 94: 'of patients' should be added after '25%'.
